# A Gedunin-Type Limonoid, 7-Deacetoxy-7-Oxogedunin, from Andiroba (*Carapa guianensis* Aublet) Reduced Intracellular Triglyceride Content and Enhanced Autophagy in HepG2 Cells

**DOI:** 10.3390/ijms232113141

**Published:** 2022-10-28

**Authors:** Akifumi Nagatomo, Kiyofumi Ninomiya, Shinsuke Marumoto, Chie Sakai, Shuta Watanabe, Wakana Ishikawa, Yoshiaki Manse, Takashi Kikuchi, Takeshi Yamada, Reiko Tanaka, Osamu Muraoka, Toshio Morikawa

**Affiliations:** 1Pharmaceutical Research and Technology Institute, Kindai University, 3-4-1 Kowakae, Higashiosaka 577-8502, Osaka, Japan; 2School of Pharmacy, Shujitsu University, 1-6-1 Nishigawara, Naka-ku, Okayama 703-8516, Okayama, Japan; 3Joint Research Center, Kindai University, 3-4-1 Kowakae, Higashiosaka 577-8502, Osaka, Japan; 4Faculty of Pharmacy, Osaka Medical and Pharmaceutical University, 4-20-1 Nasahara, Takatsuki 569-1094, Osaka, Japan; 5Faculty of Pharmacy, Toho University, 2-2-1 Miyama, Funabashi 274-8510, Chiba, Japan; 6Antiaging Center, Kindai University, 3-4-1 Kowakae, Higashiosaka 577-8502, Osaka, Japan

**Keywords:** *Carapa guianensis*, Andiroba, limonoid, gedunin, fatty liver, autophagy

## Abstract

The seed oil of *Carapa guianensis* Aublet (Andiroba) has been used in folk medicine for its insect-repelling, anti-inflammatory, and anti-malarial activities. This study aimed to examine the triglyceride (TG) reducing effects of *C. guianensis*-derived limonoids or other commercially available limonoids in human hepatoblastoma HepG2 cells and evaluate the expression of lipid metabolism or autophagy-related proteins by treatment with 7-deacetoxy-7-oxogedunin (DAOG; **1**), a principal limonoid of *C. guianensis*. The gedunin-type limonoids, such as DAOG (% of control at 20 μM: 70.9 ± 0.9%), gedunin (**2**, 74.0 ± 1.1%), epoxyazadiradione (**4**, 73.4 ± 2.0%), 17β-hydroxyazadiradione (**5**, 79.9 ± 0.6%), 7-deacetoxy-7α-hydroxygedunin (**6**, 61.0 ± 1.2%), andirolide H (**7**, 87.4 ± 2.2%), and 6α-hydroxygedunin (**8**, 84.5 ± 1.1%), were observed to reduce the TG content at lower concentrations than berberine chloride (BBR, a positive control, 84.1 ± 0.3% at 30 μM) in HepG2 cells pretreated with high glucose and oleic acid. Andirobin-, obacunol-, nimbin-, and salannin-type limonoids showed no effect on the intracellular TG content in HepG2 cells. The TG-reducing effect of DAOG was attenuated by the concomitant use of compound C (dorsomorphin), an AMPK inhibitor. Further investigation on the detailed mechanism of action of DAOG at non-cytotoxic concentrations revealed that the expressions of autophagy-related proteins, LC3 and p62, were upregulated by treatment with DAOG. These findings suggested that gedunin-type limonoids from Andiroba could ameliorate fatty liver, and that the action of DAOG in particular is mediated by autophagy.

## 1. Introduction

Highly oxygenated tetranortriterpenoids, such as limonoids, are structurally formed by the loss of four terminal carbons from the side chain in apotirucallane or apoeuphane skeleton, followed by cyclization to form the 17-furan ring [1,2,3]. Compounds belonging to this group exhibit a diverse range of biological functions, such as insecticidal, insect antifeedant, and growth-regulating activities in insects, as well as anti-bacterial, anti-fungal, anti-malarial, anti-cancer, anti-viral, and several other pharmacological activities in humans [1,2,4]. Limonoids are mainly found in the Meliaceae and Rutaceae families and less frequently in the Cneoraceae and *Harrisonia* sp. of Simaroubaceae family [1,2,3,4]. A Meliaceae plant, *Carapa guianensis* Aublet (local name Andiroba), is distributed in the tropical rainforests of countries, such as Brazil and Colombia. The seeds, flowers, and bark of this plant have been used as traditional medicines due to its insect-repelling, anti-inflammatory, and anti-malarial properties. The Andiroba seed oil has been reported to exhibit highly efficient analgesic, anti-bacterial, anti-inflammatory, anti-cancerous, anti-tumor, anti-fungal, and anti-allergic properties. In addition, it was found to be effective against wounds, bruises, herpes ulcers, rheumatism, ear infections, and insect bites as a repellent [5]. During characterization studies on the biological activities of constituents from the seed and flower oils of *C. guianensis*, we reported their cytotoxic [6,7,8,9,10], anti-malarial [11], anti-inflammatory [12,13,14,15], hepatoprotective [16], and collagen synthesis-promoting activities [17].

Non-alcoholic fatty liver disease (NAFLD) is induced by changes in diet and other lifestyle habits and is characterized by excessive triglyceride (TG) accumulation in hepatocytes, which is often associated with obesity and diabetes. As NAFLD progresses, it develops into non-alcoholic steatohepatitis (NASH) accompanied by inflammation, necrosis, or fibrosis of hepatocytes, often leading to hepatocirrhosis or hepatocellular carcinoma [18,19]. Additionally, NAFLD is closely associated with several other diseases, including chronic kidney disease, cardiovascular disorder, thyroid disease, polycystic ovary syndrome (PCOS), and colon cancer [20,21,22,23,24,25]. The first line of treatment for NAFLD/NASH coexisting with obesity as defined by the ESPEN guideline is intensive lifestyle intervention leading to weight loss and exercise leading to hepatic fat reduction [26]. Reports documenting effective NAFLD/NASH treatment include pioglitazone, a sodium-glucose cotransporter 2 inhibitor, or glucagon-like peptide-1 receptor agonist for type 2 diabetes [27,28,29], statins for hypercholesterolemia [30], angiotensin II receptor blockers and angiotensin II-converting enzyme inhibitors for hypertension [31,32], high-dose ursodeoxycholic acid [33], the anti-oxidant vitamin E, [34] and obeticholic acid [35], a nuclear receptor FXR agonist acting as bile acid. However, no specific medicine has been approved; therefore, the development of a NAFLD/NASH-specific medicine is essentially required [36].

We have been searching for constituents from natural products that can contribute to the prevention and improvement of NAFLD or any other lifestyle-related disease in vivo models using mice or in cell-based assay using HepG2 cells. Studies on hepatic triglyceride metabolism often use HepG2 cells and other cell lines derived from liver, as well as liver slices or primary hepatocytes [37]. We identified several flavonoids [38,39,40], megastigmanes [41], and diterpenes [42,43,44] with anti-fatty liver principles, which showed inhibitory effects on TG accumulation or accelerated TG metabolism in hepatocytes. In addition, sesquiterpene glycosides were obtained from artichoke (the leaves of *Cynara scolymus* L.) [45], saponins from tea flower buds of *Camellia sinensis* (L.) Kuntze [46,47], daisy flowers of *Bellis perennis* L. [48,49], pericarps of *Sapindus rarak* DC. [50], mate leaves of *Ilex paraguariensis* A. St.-Hil. [51], and oligostilbenes from the bark of *Shorea roxburghii* G. Don [52] showed anti-hyperlipidemic effects in olive oil-treated mice. In the previous report, we elucidated that limonoids isolated from Andiroba strongly reduced the intracellular TG content in high glucose-pretreated HepG2 cells [53]. This study examined the relationship between the different types of limonoids and their TG-reducing activity against accumulated TG in HepG2 cells concomitantly pretreated with high glucose and oleic acid. Furthermore, the mechanism of action of 7-deacetoxy-7-oxogedunin (DAOG; **1**), a principal limonoid constituent of Andiroba, was investigated to study the involvement of autophagy in TG-reducing activity.

## 2. Results

### 2.1. Effect of Limonoids on TG Content in HepG2 Cells

HepG2 cells treated with high concentrations of glucose or fatty acids have been reported to cause intracellular TG accumulation [37]. Here, HepG2 cells were incubated for 24 h in a medium containing high concentrations of glucose with oleic acid to induce intracellular TG accumulation and were evaluated for intracellular TG content post-24 h by adding 14 different limonoids (**1**–**14**, Figure 1). Among the limonoids used in this study, gedunin- (**1**–**8**) and andirobin-type (**9**) limonoids were isolated from Andiroba, while the others were commercially available products. Obacunol-type limonoids are abundant in citrus fruits; nomilin (**10**) and obacunone (**11**) constitute the biosynthetic pathway of limonin (**12**) [54]. Nimbin (**13**, nimbin-type) and nimbolide (**14**, salannin-type) are principal limonoids found in *Azadirachta indica* [2]. All the gedunin-type limonoids expressed TG-reducing activity except 6α-acetoxygedunin (**3**, % of control at 20 μM: 101.9 ± 0.9%), with 7-deacetoxy-7-oxogedunin (DAOG; **1**, 70.9 ± 0.9%) and 7-deacetoxy-7α-hydroxygedunin (**6**, 61.0 ± 1.2%) being the most active compounds (Table 1). Other types of limonoids showed weak activity (**11**, 87.7 ± 1.9% at 20 μM) or were inactive. Compound **14** reduced TG at low concentrations (74.6 ± 0.8% at 5 μM) but was cytotoxic at 20 μM. We used berberine chloride (BBR) as a positive control, a benzylisoquinoline alkaloid widely found in a variety of plants, including *Coptis japonica* (Thunb.) Makino (Ranunculaceae) and *Phellodendron amurense* Rupr. (Rutaceae) [55]. *Trans*-tiliroside, an acylated flavonol glycoside, has been reported to reduce intracellular TG content in high glucose-pretreated HepG2 cells [39], and it showed the same activity in the present experimental conditions. The lipid droplets formed by the intracellularly accumulated TG were stained using Oil red O, as shown in Figure 2. Intracellular lipid droplets are rarely seen in a normal medium; however, in a medium with high glucose and fatty acid concentrations, numerous lipid droplets with accumulated TG appear intracellularly. The addition of DAOG (**1**, 20 μM) reduced the number of intracellular lipid droplets (Figure 2, right panel) in HepG2 cells.

### 2.2. Involvement of Adenosine Monophosphate-Activated Protein Kinase (AMPK) in the Mechanism of Action of DAOG (**1**)

#### 2.2.1. Cell Viability

3-(4,5-Dimethyl-2-thiazolyl)-2,5-diphenyltetrazolium bromide (MTT) assay was performed to confirm the effects of DAOG (**1**) and BBR on HepG2 cell viability and to determine their optimal concentrations. Post-intracellular TG accumulation using high glucose and oleic acid-containing Dulbecco’s Modified Eagle’s Medium (DMEM), the test samples were applied for 24 h. The cell viability decreased in the presence of DAOG (**1**) at concentrations of more than 10 μM (96.2 ± 0.5%), with no change in appearance (Figure 3). On the contrary, the cells were almost entirely killed in the presence of 100 µM BBR (1.6 ± 0.7%), while BBR at a concentration of 30 µM (96.2 ± 0.4%) showed little effects (all of numerical data described in Appendix A). Accordingly, 20 µM and 30 µM were set as the upper concentrations for DAOG (**1**) and BBR, respectively, for further studies.

#### 2.2.2. Effects of AMPK Inhibitor on Intracellular TG Content as Well as AMPK and ACC Phosphorylation

To investigate the AMPK pathway involvement in the intracellular TG-reducing effect of DAOG (**1**), the combined effect of a potent AMPK inhibitor, compound C [56], was confirmed. Compound **1** (% of control: 71.3 ± 1.6% at 20 μM) and BBR (82.6 ± 2.4% at 30 μM) showed a concentration-dependent reduction of TG accumulated intracellularly using high glucose and oleic acid concentrations, as indicated by the blank bars in Figure 4A. In contrast, the activity of DAOG (**1**) and BBR was significantly attenuated by the concomitant use of compound C (Figure 4A, filled bars, 97.9 ± 0.9% at 20 μM of **1**, 149.0 ± 2.6% at 30 μM of BBR), as fully described in Appendix A. The attenuation of the TG-reducing effect by inhibiting AMPK with compound C was more effective in combination with BBR. Our results are consistent with previous studies, reporting a beneficial effect of BBR on NAFLD through the AMPK pathway [57,58].

Next, the expression of phosphorylated AMPK (the active form of AMPK) was analyzed by immunoblotting. Compound **1** or BBR was added to HepG2 cells in which TG accumulation was induced using high glucose and oleic acid concentrations; post-24 h, the cells were collected, and the proteins were extracted for western blotting. Figure 4B,C indicate that BBR markedly increased the p-AMPK/AMPK ratio (3.21 ± 0.66-fold increase, Appendix A). In addition, compound **1** also tended to increase the p-AMPK/AMPK ratio (1.43 ± 0.07-fold increase, Appendix A), however, the change was insignificant, and relative to the control (only vehicle-treated). Acetyl-CoA carboxylase (ACC) is a key molecule in fatty acid metabolism and is negatively regulated by AMPK. ACC catalyzes the conversion of acetyl-CoA to malonyl-CoA, which inhibits carnitine palmitoyltransferase (CPT)-1. Because CPT-1 catalyzes the transfer of the fatty acid groups from acyl-CoA to carnitine and increases their transport from cytosol to mitochondria, inactivation of ACC by AMPK enhances fatty acid degradation via β-oxidation [59]. In our study, ACC phosphorylation was not clearly observed despite AMPK activation (0.86 ± 0.09-fold increase for **1**, 0.96 ± 0.14-fold increase for BBR, Appendix A). However, the p-ACC/ACC ratio significantly decreased due to treatment with compound C (Appendix A).

### 2.3. Involvement of Autophagy in the Mechanism of Action of DAOG (**1**)

In Section 2.2, the lipid-reducing effect of DAOG (**1**) was implicated to be mediated by AMPK; however, the western blot analysis results suggested that the enhancement of β-oxidation is not very strongly induced through the AMPK/ACC pathway. Therefore, we investigated the effects of compound **1** and its involvement in lipophagy, the selective breakdown of droplet-stored lipids by autophagy. Time-dependent expression of several autophagy-related proteins, such as microtubule-associated protein 1 light chain 3 (MAP1LC3, also known as LC3), sequestosome 1 (SQSTM1/p62), and Run domain Beclin-1 interacting and cysteine-rich containing protein (Rubicon), was analyzed by western blotting (Figure 5A,B). Compound **1** stimulated the conversion of LC3-I into LC3-II, and the LC3-II expression at 24 h after the addition of compound **1** (1.55 ± 0.17-fold increase, Appendix A) was significantly increased compared to the initial value. SQSTM1/p62 is an autophagy adaptor that directly binds to LC3 to induce autophagosome formation [60]. Typically, SQSTM1/p62 is degraded during autophagy, but our results showed that compound **1** caused a rapid increase in SQSTM1/p62 protein expression (2.74 ± 0.15-fold increase at 6 h, Appendix A), followed by a decrease over time. Meanwhile, the expression of Rubicon, a negative regulator of autophagy [61], was observed to decrease in response to treatment with compound **1** (0.62 ± 0.08-fold increase at 24 h, Appendix A). Since the accumulation of LC3-II and SQSTM1/p62 is also caused by inhibition of autophagy, an autophagy flux assay was performed to determine whether the present results obtained with compound **1** were due to enhancement or inhibition of autophagy. Chloroquine (CQ) inhibits autophagy at its final stage by suppressing the fusion of autophagosomes with lysosomes [62]. The results in Figure 5C showed that LC3-II expression in the control group was significantly increased in the presence of CQ (1.84 ± 0.05-fold increase, Appendix A); however, an even more significant increase was observed with the combination of compound **1** and CQ (2.58 ± 0.21-fold increase, Appendix A). Furthermore, cells were collected 4 h after sample addition, and mRNA expression was confirmed by real-time quantitative reverse transcription polymerase chain reaction (qRT-PCR); the mRNA expression levels of *SQSTM1* and *RUBCN* were observed to increase and decrease, respectively, compared to the control (relative mRNA level was 2.93 ± 0.07 for *SQSTM1*, 0.63 ± 0.02 for *RUBCN*, Appendix A). This indicated that compound **1** enhanced autophagy in HepG2 cells pretreated with high glucose and oleic acid concentrations.

## 3. Discussion

Disruption of hepatic lipid homeostasis resulting in hepatic TG accumulation is commonly described based on five concepts: (1) increased fatty acid influx, (2) increased de novo lipogenesis, (3) reduced TG secretion, (4) reduced fatty acid oxidation, and (5) dysfunctional autophagy [60,63,64]. As a research strategy to find naturally occurring ingredients that contribute to the prevention and improvement of fatty liver (NAFLD) associated with lifestyle-related diseases, we constructed a screening system to induce intracellular TG accumulation using specific culture medium and screened various herbal medicines and food-derived ingredients, with an increase or decrease in intracellular TG content as indicators. In our study, HepG2 cells were cultured in DMEM containing high glucose with oleic acid. After 24 h, the amount of intracellular TG content greatly increased compared to the control. The cells were treated with several limonoids, and most of the gedunin-type limonoids isolated from Andiroba reduced intracellular TG content by 10–40% compared to the control at 24 h (Table 1). Gedunin-type limonoids (**1**–**8**) are intact ABCD limonoid ring skeleton or the D-ring-*seco* limonoids, while compound **9** is the B,D-ring-*seco* limonoid, compounds **10**–**12** are the A,D-ring-*seco*-limonoids, and compounds **13** and **14** are members of the C-ring-*seco* class (Figure 1). Therefore, the intact A, B, and C-rings of limonoid may be essential for the activity expression, however, further evaluation of wider range of limonoids is required. In the gedunin-type limonoids, 7-deacetoxy-7α-hydroxygedunin (**6**) with a hydroxy group at C-7 position showed the strongest activity, while the activity vanished in the case of 6α-acetoxygedunin (**3**), with two acetoxy groups at C-6 and C-7 positions. The activity was enhanced by the presence of an oxo-substituent at C-7 position. This indicates that the substitutions at these positions in the gedunin skeleton affected the activity.

There have been several reports on the anti-obesity effects of limonoids; nomilin (**10**) and obacunone (**11**) are agonists of the bile acid-specific G protein-coupled receptor, TGR5, and are known to improve insulin resistance and increase energy expenditure by activating TGR5 in L cells of the small intestine and skeletal muscle [65,66,67]. In addition, nimbolide (**14**) has exhibited anti-obesity effects by reducing oxidative stress in previous studies using primary cultured hepatocytes and high-fat diet-loaded rats [68,69]. Ceramicine B derived from *Chisocheton ceramicus* has been reported to suppress adipocyte differentiation and lipid-droplet accumulation due to interruption of the phosphorylation of Forkhead box O1 (Foxo1) in the mouse preadipocyte cell line MC3T3-G2/PA6 [70]. Among the Andiroba-derived limonoids, DAOG (**1**) has been reported to enhance the brown adipose tissue to improve hyperlipidemia and insulin resistance in diet-induced obese mice and suppress lipid accumulation by reducing glucose transporter (GLUT) 4 expression in adipocytes via insulin receptor substrate (IRS)-1/Akt axis [71,72]. However, the involvement of AMPK in the anti-obesity mechanism of limonoids has not been described clearly. Therefore, in the present study, we investigated the involvement of AMPK in the mechanism of action of gedunin-type limonoids, particularly compound **1**, in reducing the effects of intracellular TG.

AMPK is the key metabolic sensor for cellular energy status and is a serine/threonine protein kinase that plays a central role in regulating cellular metabolism and energy balance in mammalian cells. AMPK is activated by changes in intracellular levels of AMP:ATP and ADP:ATP ratios, or by upstream kinases, such as liver-kinase-B1 or calcium/calmodulin-dependent kinase 2, also known as CAMKKβ [73]. Activated AMPK (phosphorylation at Thr172 in the AMPKα subunit) induces ATP-generating pathways, such as glycolysis, fatty acid oxidation, and lipolysis, while the anabolic pathways, such as gluconeogenesis, cholesterol, and protein synthesis, are inhibited. Therefore, AMPK acts comprehensively on lipid metabolism by promoting fatty acid oxidation and inhibiting cholesterol and TG synthesis [73]. AMPK has attracted attention as a therapeutic target for various diseases, and many naturally occurring compounds have shown the ability to activate AMPK, including BBR or other constituents belonging to the polyphenol family [74]. BBR has several beneficial functions, such as antioxidant [75,76,77], cardiovascular protective [78,79], anti-hyperlipidemic [80], anti-hyperglycemic [81], hepatoprotective [82,83], nephroprotective [84,85], and immunomodulatory effects [86,87]. BBR has been reported to attenuate NAFLD in a clinical study [88] and inhibit TG levels in steatotic hepatocytes [89]. Our results indicated that both compound **1** and BBR, in combination with the AMPK inhibitor compound C, exhibit AMPK-mediated intracellular TG-reducing effects (Figure 4A). However, western blot analysis showed that the AMPK activation effect of compound **1** was less prominent than that of BBR (Figure 4B,C). In addition, no obvious change in ACC phosphorylation (inactivation) associated with AMPK activation was observed, suggesting that further optimization of experimental conditions, such as the cell collection timing, is necessary. Furthermore, compound **1** have been reported to repress the intracellular TG accumulation in adipocytes [72]. Since AMPK activation also suppresses de novo lipogenesis, it is consistent with our result of compound **1** activating AMPK. Thus, we presume that compound **1** regulates lipid metabolism via AMPK.

Autophagy is an evolutionarily conserved cytoplasmic degradation system that controls autolysosomal degradation and recycling of cellular components. Lipophagy is a specific form of autophagy that mediates the selective breakdown of lipid droplets in the liver and enhances β-oxidation [63,64,90,91]. Disturbances in lipophagy have been reported to be relevant to NAFLD and hepatic steatosis [92,93,94]. Autophagy is mainly regulated by mechanistic targeting of rapamycin complex 1 (mTORC1), which is responsible for anabolism and is activated by starvation. In addition, the activated AMPK promotes autophagy directly by phosphorylating autophagy-related proteins, such as mTORC1 and unc-51 like autophagy activating kinase 1 (ULK1) [95,96]. Hence, AMPK activation by compound **1** could affect intracellular TG content through the enhancement of autophagy.

Our results showed that compound **1** increased the expression of LC3-II over time and transiently increased p62 (Figure 5A,B). LC3-I is cleaved from LC3 by autophagy-related gene 4, which binds to phosphatidylethanolamine (PE) to become LC3-II and participates in the elongation of isolation membranes [91]. Therefore, the expression level of LC3-II is frequently used as a marker of autophagy activation. However, since increased levels of LC3-II analyzed by immunoblotting may result from dysfunction of the fusion process with lysosomes, an increase in LC3-II alone cannot be considered to determine autophagy enhancement [96]. Therefore, we confirmed alterations in LC3-II expression through an autophagy flux assay in which an autophagy inhibitor artificially induces a dysfunctional state of autophagy. The results showed that LC3-II accumulation by CQ in the presence of compound **1** was significantly higher than that by CQ alone (Figure 5C). This indicates that compound **1** could enhance autophagy.

SQSTM1/p62 is a receptor protein that directs ubiquitinated autophagy substrate proteins to the isolation membrane, where they interact directly with LC3 and are degraded in an autophagy-dependent manner [97,98,99]. SQSTM1/p62 also accumulates in cells due to inhibition of autophagy; however, compound **1** increased SQSTM1/p62 mRNA expression transiently, which declined gradually. Evaluation of SQSTM1/p62 mRNA expression 4 h after the addition of compound **1** revealed that it was significantly higher than that in controls (Figure 5D). This suggests that compound **1** stimulates SQSTM1/p62 expression and enhances autophagy. Although increased SQSTM1/p62 expression has been reported to trigger the enhancement of autophagy [100,101,102], further investigation is needed to determine whether a similar mechanism applies to compound **1**. Furthermore, compound **1** decreased the protein and mRNA expression levels of Rubicon (Figure 5B,D), an inhibitor of autophagy that has been reported to be associated with NAFLD because of its increased concentration in the liver of high-fat diet-loaded mice, resulting in inhibition of lipid droplet degradation [61]. In addition, Rubicon is reported as one of the signatures of aging because autophagic dysfunction with age in several organisms is induced by an age-dependent increase in Rubicon [103]. The reduction of Rubicon by compound **1** supports autophagy activation; however, a decrease in mRNA expression suggests that compound **1** may also be useful in aging caused by autophagy deficiency (Figure 5B,D). Few reports have shown that nimbolide (**14**) and azadirachtin induce autophagy [104,105], while gedunin (**2**) is reported to downregulate it [106], which is indicative of the relationship between limonoids and autophagy. However, our study is the first report demonstrating that limonoids improve intracellular lipid metabolism through autophagy. Presently, only gedunin-type limonoids have been studied, and more research data should be accumulated in the future.

In the current study, only in vitro evaluation using cells was conducted. The pharmacokinetics of limonoids have been well studied, especially for limonin (**12**). Generally, the bioavailability of limonin is very low due to its low solubility and poor permeability [107,108]. Although there are no reports on the pharmacokinetics of compound **1**, it is also sparingly soluble in water, so the bioavailability is thought to be low. However, Matsumoto et al. elucidated the anti-obesity and the anti-diabetic effects of DAOG (**1**) by an in vivo model using high-fat-diet-fed mice [71]. Thus, compound **1** is expected to exhibit oral absorbability to some extent.

## 4. Materials and Methods

### 4.1. General

The following instruments were used to obtain spectroscopic data: specific rotation, JASCO P-2200 polarimeter (JASCO Corporation, Tokyo, Japan, *l * =  5 cm); UV spectra, Shimadzu UV-1600 spectrometer; IR spectra, IRAffinity-1 spectrophotometer (Shimadzu Co., Kyoto, Japan); ^1^H NMR spectra, JNM-ECA800 (800 MHz), and JNM-ECS400 (400 MHz) spectrometers; ^13^C NMR spectra, JNM-ECS400 (100 MHz) spectrometer (JEOL Ltd., Tokyo, Japan). Determinations were made using samples dissolved in deuterated chloroform at room temperature (RT) with tetramethylsilane as an internal standard; ESI-MS and high-resolution ESI-MS, Exactive™ Plus Orbitrap mass spectrometer (Thermo Fisher Scientific Inc., Waltham, MA, USA); HPLC detector, SPD-10A*vp* UV-VIS detector; HPLC columns, Cosmosil 5C_18_-MS-II (Nacalai Tesque, Inc., Kyoto, Japan). Columns of 4.6 mm i.d. × 250 mm and 20 mm i.d. × 250 mm were used for analytical and preparative purposes, respectively.

The following chromatographic materials were used for column chromatography (CC) experiments: normal-phase silica gel CC, silica gel 60 N (Kanto Chemical Co., Ltd., Tokyo, Japan; 63–210 mesh, spherical, neutral); reversed-phase ODS CC, Chromatorex ODS DM1020T (Fuji Silysia Chemical, Ltd., Aichi, Japan; 100–200 mesh); thin-layer chromatography (TLC), precoated TLC plates with silica gel 60F_254_ (Merck, Darmstadt, Germany, 0.25 mm) (normal-phase) and silica gel RP-18 WF_254S_ (Merck, 0.25 mm) (reversed-phase); reversed-phase HPTLC, precoated TLC plates with silica gel RP-18 WF_254S_ (Merck, 0.25 mm). Detection was carried out by spraying with 1% Ce(SO_4_)_2_–10% aqueous H_2_SO_4_ followed by heating.

### 4.2. Plant Material

The flower and seed oils of *C. guianensis* Aublet (Meliaceae) were collected in Amazon, Brazil, in March of 2006, 2011, and 2013. Voucher specimens (CG-01-1, CGS-01-1, and CGS-01-2) were deposited at the Herbarium of the Laboratory of Medicinal Chemistry, Osaka University of Pharmaceutical Sciences, as described previously [16].

### 4.3. Isolation of Limonoids

The Andiroba oil (252.1 g) [6,7,11,12] was subjected to normal-phase silica gel CC (3.0 kg, CHCl_3_ → EtOAc → MeOH) to yield 12 fractions (Fr. 1 (147.9 g), Fr. 2 (70.6 g), Fr. 3 (21.4 g), Fr. 4 (8.5 g), Fr. 5 (1.8 g)). Fraction 3 (21.4 g) was subjected to reversed-phase ODS CC (640.0 g, MeOH:H2O (60:40 → 80:20 → 100:0, *v*/*v*) → acetone) to yield 10 fractions (Fr. 3-1 (0.2 g), Fr. 3-2 (0.5 g), Fr. 3-3 (1.8 g), Fr. 3-4 (67.7 mg), Fr. 3-5 (0.6 g), Fr. 3-6 (0.3 g), Fr. 3-7 (1.5 g), Fr. 3-8 (7.3 g), Fr. 3-9 (9.9 g), Fr. 3-10 (0.7 g)). The fraction 3-3 (200.0 mg) was purified by HPLC (detection: UV (230 nm), column: Cosmosil 5C_18_-MS-II, CH_3_CN–H_2_O (50:50, *v*/*v*)) to obtain 6α-acetoxygedunin (**3**, 21.3 mg, 0.0761%) [109], andirolide H (**7**, 11.5 mg, 0.0411%) [11], and methyl angolensate (**9**, 36.8 mg, 0.1316%) [109]. Likewise, fraction 3-4 (67.7 mg) was purified by HPLC (conditions similar to fraction 3-3, except 20% CH_3_CN) to obtain 6α-acetoxygedunin (**3**, 12.6 mg, 0.0050%) [109] and 7-deacetoxy-7α-hydroxygedunin (**6**, 10.4 mg, 0.0041%) [110]. The fraction 3-5 (200.0 mg) was purified by HPLC (detection: UV (230 nm), column: Cosmosil 5C_18_-MS-II, CH_3_CN–H_2_O (50:50, *v*/*v*)) to obtain DAOG (**1**, 28.0 mg, 0.0317%) [109], gedunin (**2**, 44.8 mg, 0.0506%) [111], and 7-deacetoxy-7α-hydroxygedunin (**6**, 25.2 mg, 0.0285%) [110]. Each isolate was identified by comparing various physical and NMR (^1^H and ^13^C) data with previous reports. Other limonoids derived from Andiroba (**4**, **5**, and **8**) were previously isolated compounds [16]. ^1^H- and ^13^C-NMR spectrum of DAOG (**1**) are shown in Appendix A.

### 4.4. Reagents

Limonin (**12**) was purchased from Tokyo Chemical Industry Co., Ltd. (Tokyo, Japan); nomilin (**10**) and nimbin (**13**) were procured from Funakoshi Co., Ltd. (Tokyo, Japan); obacunone (**11**) and nimbolide (**14**) were obtained from Sigma-Aldrich Co. LLC. (St. Louis, MO, USA). Other chemicals, unless otherwise indicated, were purchased from FUJIFILM Wako Pure Chemical Corporation (Tokyo, Japan).

### 4.5. Cell Culture

HepG2 cells obtained from RIKEN Cell Bank (RCB1648, Tsukuba, Japan) were maintained in Minimum Essential Medium Eagle (MEM, FUJIFILM Wako) containing 10% fetal bovine serum, 1% MEM non-essential amino acids (FUJIFILM Wako), penicillin G (100 units/mL), and streptomycin (100 μg/mL) at 37 °C under 5% CO_2_ atmosphere.

### 4.6. Effect on TG Content in HepG2 Cells Pretreated with High Glucose and Oleic Acid

HepG2 cells were inoculated into a 48-well tissue culture plate (10^5^ cells/well in 150 μL/well MEM). After 24 h of incubation, the medium was replaced with 150 μL/well of DMEM (FUJIFILM Wako) containing low glucose (1000 mg/L), and the cells were cultured until 90% confluency. The medium was then replaced with DMEM containing high glucose (4500 mg/L) and 5% (*v*/*v*) oleic acid albumin (Sigma) for induction of intracellular TG accumulation. After 24 h, the medium was replaced with low glucose-DMEM containing a test sample or vehicle, and the cells were cultured for another 24 h. Subsequently, the medium was removed, and the cells were homogenized in distilled water (105 μL/well) by sonication. The TG and protein content in the homogenate were determined using commercial kits (LabAssay™ Triglyceride and Protein Assay BCA Kit, FUJIFILM Wako); data were expressed as the % of control of TG/protein (μg/mg). Each test compound was dissolved in dimethyl sulfoxide (DMSO) and added to the medium (final DMSO concentration was up to 1%). Compound C was applied in combination with the sample.

### 4.7. Oil Red O Staining

After 24 h of the sample application in TG-accumulated HepG2 cells, the cells were washed twice with phosphate-buffered saline (PBS) (−), followed by fixation using 4% neutral-buffered formalin for 10 min. The cells were again washed by PBS (−) two times and stained with 1.8 mg/mL Oil red O in 60% isopropanol for 30 min. Following staining and washing with distilled water twice, the cells were observed by microscopy.

### 4.8. Cell Viability

MTT assay was performed for cell viability determination. Briefly, MTT solution (5 mg/mL in PBS (−)) was diluted using DMEM (final MTT concentration: 0.5 mg/mL). After 24 h of sample treatment in HepG2 cells, the DMEM was replaced with MTT-DMEM and incubated for an additional 2 h. The medium was then removed, and 0.04 N HCl in isopropanol (100 μL/well) was added, followed by 30 min of incubation at RT to dissolve the formazan. The absorbance was measured on a microplate reader (Multiskan Sky, Thermo Fisher Scientific) using a test wavelength of 560 nm and a reference wavelength of 670 nm. Cell viability was determined using the following equation:Cell viability (%) = *A/B* × 100,
where *A* is the absorbance with the test sample and *B* is the absorbance with the control.

### 4.9. Western Blot Analysis

After induction of TG accumulation and the sample application, as described in Section 4.6, protein extraction was performed by directly dispensing M-PER™ Mammalian Protein Extraction Reagent (Thermo Fisher Scientific), containing protease and phosphatase inhibitors cocktail, into the wells at 0, 6, 12, and 24 h post-sample addition. The lysates were prepared in 4x Laemmli Sample Buffer (Bio-Rad Laboratory, Hercules, CA, USA) containing 2-mercaptoethanol. Equivalent amounts of protein (20 μg) were separated on 4–20% Mini-PROTEAN^®^ TGX™ Precast Protein Gels (Bio-Rad) and transferred onto polyvinylidene fluoride membranes. After blocking the membranes with Blocking One-P (Nacalai Tesque) for 1 h, the target antigen was allowed to react with the primary antibodies at 4 °C overnight. All the antibodies were purchased from Cell Signaling Technology, Inc. (Danvers, MA, USA). The antibodies and dilution ratios are as follows; ACC (Acetyl-CoA Carboxylase Antibody #3662, 1:1000), p-ACC (Phospho-Acetyl-CoA Carboxylase (Ser79) (D7D11) Rabbit mAb #11818, 1:1000), AMPK (AMPKα (D5A2) Rabbit mAb #5831, 1:1000), p-AMPK (Phospho-AMPKα (Thr172) Rabbit mAb #2535, 1:1000), LC3-I/II (LC3A/B (D3U4C) XP^®^ Rabbit mAb #12741, 1:1000), SQSTM1/p62 (SQSTM1/p62 Antibody #5114, 1:1000), Rubicon (Rubicon (D9F7) Rabbit mAb #8465, 1:1000), and β-actin (β-Actin (13E5) Rabbit mAb (HRP Conjugate) #5125, 1:2000). The membranes were then incubated with an anti-rabbit horseradish peroxidase-conjugated secondary antibody (1:5000) for 1 h at RT. The immunoreactive bands were detected using the Immunostar^®^ Zeta (FUJIFILM Wako), and the band intensity was measured using CS Analyzer (Version 3.00.1011, ATTO Corporation, Tokyo, Japan).

### 4.10. Autophagy Flux Assay

The induction of TG accumulation in HepG2 cells was performed as mentioned in Section 4.6. After 18 h of sample application, 50 μL of DMEM containing CQ (final concentration; 100 μM) was added, and the cells were further incubated for 6 h. Total protein was then extracted, followed by the determination of LC3-II and β-actin expression levels by western blot analysis. β-Actin was used as a loading control. Autophagic flux was determined by comparing differences in LC3-II expression between control and sample groups with and without CQ addition.

### 4.11. qRT-PCR

The samples were added to HepG2 cells after induction of intracellular TG accumulation using high glucose and oleic acid containing DMEM. After 4 h of sample application, the cells were collected, and the total RNA was extracted using RNeasy Mini Kit (QIAGEN, Venlo, The Netherlands) according to the manufacturer’s instructions. The concentration and purity of the RNA were determined by the absorbance ratio at 260 and 280 nm using NanoDrop Lite (Thermo Fisher Scientific). Complementary DNAs (cDNAs) were synthesized from 1 μg total RNA using iScript™ Reverse Transcription Supermix (Bio-Rad). The template cDNAs obtained were incubated with gene-specific primers and SsoAdvanced™ Universal SYBR^®^ Green Supermix (Bio-Rad) in a CFX Connect RT-PCR Detection System (Bio-Rad). The abundance of each gene product was calculated by relative quantification using the 2^−^^ΔΔCq^ method, with values for the target genes normalized to RPLP0. The thermal cycling program had an initial denaturation step (98 °C for 30 s), followed by 40 cycles of denaturation (98 °C for 15 s) and annealing/extension (60 °C for 30 s). The primer pairs used were: *SQSTM1* primers, 5′-TGCCCAGACTACGACTTGTG-3′ and 5′-AGTGTCCGTGTTTCACCTTCC-3′ [101]; *RUBCN* primers, 5′-GATTACTGGCAGTTCGTGAAAGA-3′ and 5′-CTGCTCTGGTCGTTCTCGTG-3′ [112]; and *RPLP0* primers, 5′-TCTACAACCCTGAAGTGCTTGAT-3′ and 5′-CAATCTGCAGACAGACACTGG-3′ [113].

### 4.12. Statistics

The experiments were repeated at least four times. Values are expressed as mean ± S.E.M. Student’s *t*-test and one-way analysis of variance, followed by Dunnett’s or Tukey–Kramer’s HSD test, were used for statistical analysis with JMP^®^ 9.0.2 (SAS Institute Inc., NC, USA). Probability (*p*) values less than 0.05 were considered significant.

## 5. Conclusions

Several types of limonoids, including Andiroba (*C. guianensis*)-derived limonoids, were examined for their intracellular TG-reducing effects, and we found that the gedunin-type limonoids characteristically reduced intracellular TG content. Consideration of the structure-activity relationship suggested that the substituents at the C-6 and C-7 positions in the gedunin skeleton affected the activity. DAOG (**1**), a principal limonoid of Andiroba, was observed to be involved in AMPK activation as part of its mechanism of action. In addition, compound **1** increased autophagy markers and decreased the expression of autophagy suppressors, suggesting that its action is mediated by autophagy.

## Figures and Tables

**Figure 1 ijms-23-13141-f001:**
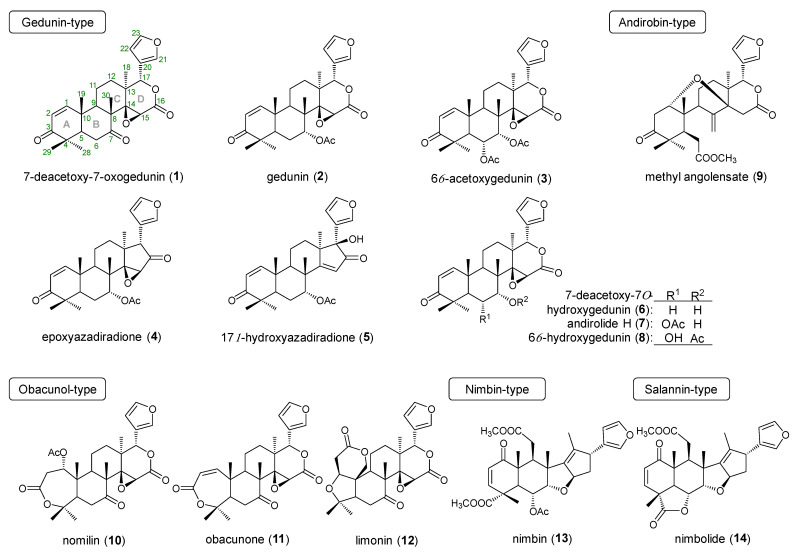
Chemical structures of limonoids.

**Figure 2 ijms-23-13141-f002:**
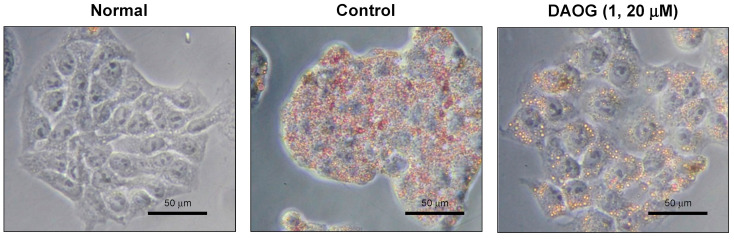
Oil red O staining of lipid droplets in HepG2 cells. Cells in the normal panel were treated with low glucose Dulbecco’s Modified Eagle Medium (DMEM) for 24 h. Reddish-colored particles are triglycerides accumulated lipid droplets in cells treated with high glucose and oleic acid. Sample or vehicle was applied for 24 h after induction of lipid accumulation.

**Figure 3 ijms-23-13141-f003:**
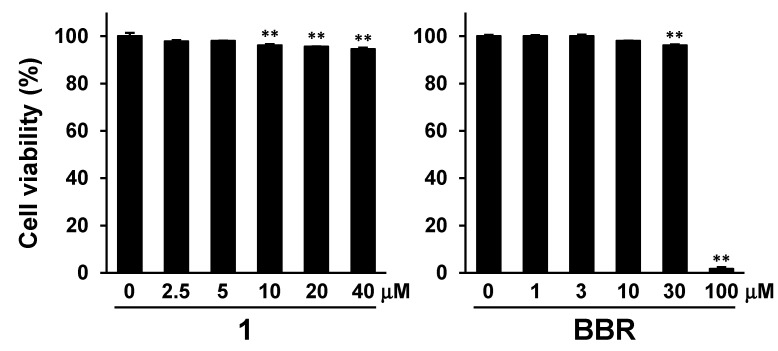
Effect of DAOG (**1**) and BBR on HepG2 cell viability by MTT assay. 7-Deacetoxy-7-oxogedunin (DAOG; **1**) and berberine chloride (BBR) were applied for 24 h after induction of intracellular triglyceride (TG) accumulation using high glucose and oleic acid. Each bar represents the mean with standard error of the mean (S.E.M.) (*n* = 4). Significantly different from the control, ** *p* < 0.01 (Dunnett).

**Figure 4 ijms-23-13141-f004:**
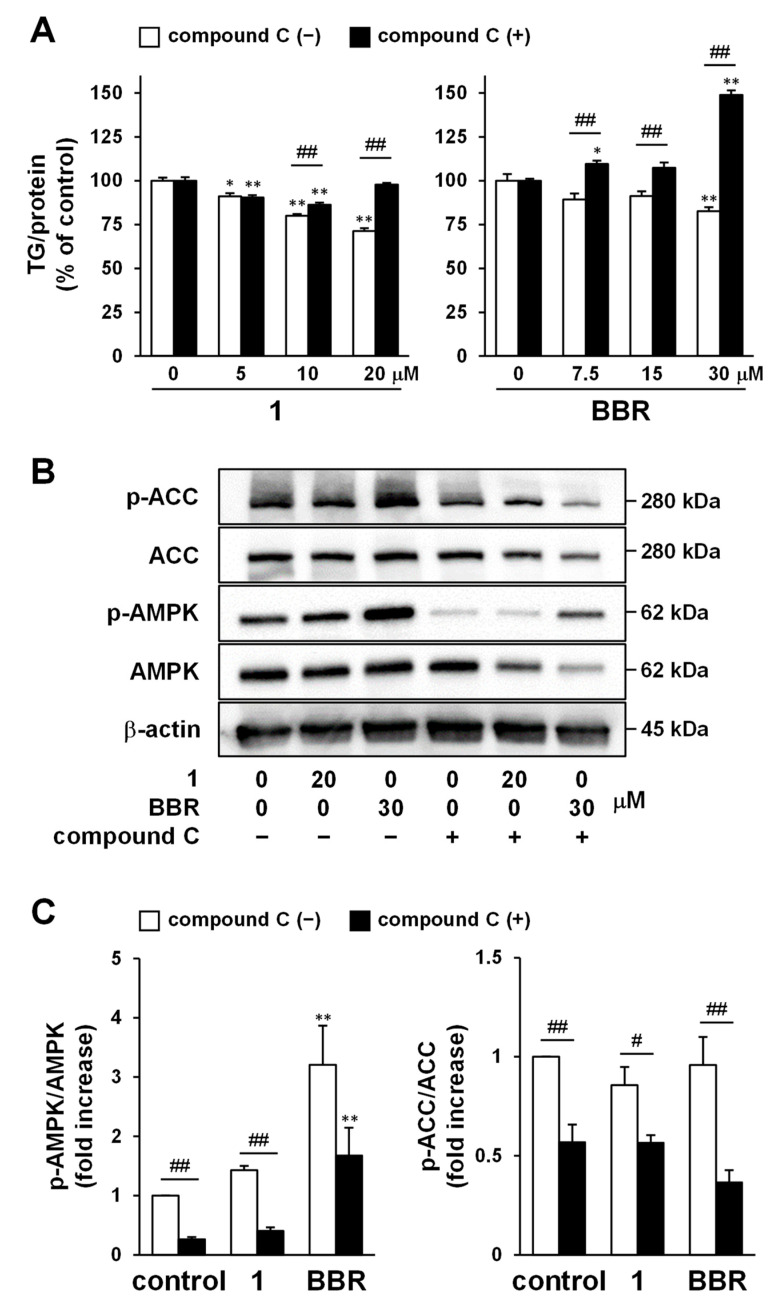
Investigation of AMPK involvement in the mechanism of intracellular TG reduction in HepG2 cells. 7-Deacetoxy-7-oxogedunin (DAOG; **1**) and berberine chloride (BBR) were applied for 24 h after induction of intracellular triglyceride (TG) accumulation by high glucose and oleic acid treatment. (**A**) Compound **1** and BBR reduced intracellular TG content in a concentration-dependent manner. The effects were abolished in the presence of a coexisting compound C (dorsomorphin); (**B**,**C**) Phosphorylation level of AMPK and acetyl-CoA carboxylase (ACC) in HepG2 cells treated with compound **1** (20 μM) or BBR (30 μM), as determined by western blot analysis. Compound C was used at a concentration of 20 μM. Unedited blots are shown in Appendix A. Each bar represents the mean with standard error of the mean (S.E.M.) (*n* = 4); * *p* < 0.05, ** *p* < 0.01 vs. control cells treated with vehicle (Dunnett); # *p* < 0.05, ## *p* < 0.01 vs. compound C-nontreated cells (Student’s *t*).

**Figure 5 ijms-23-13141-f005:**
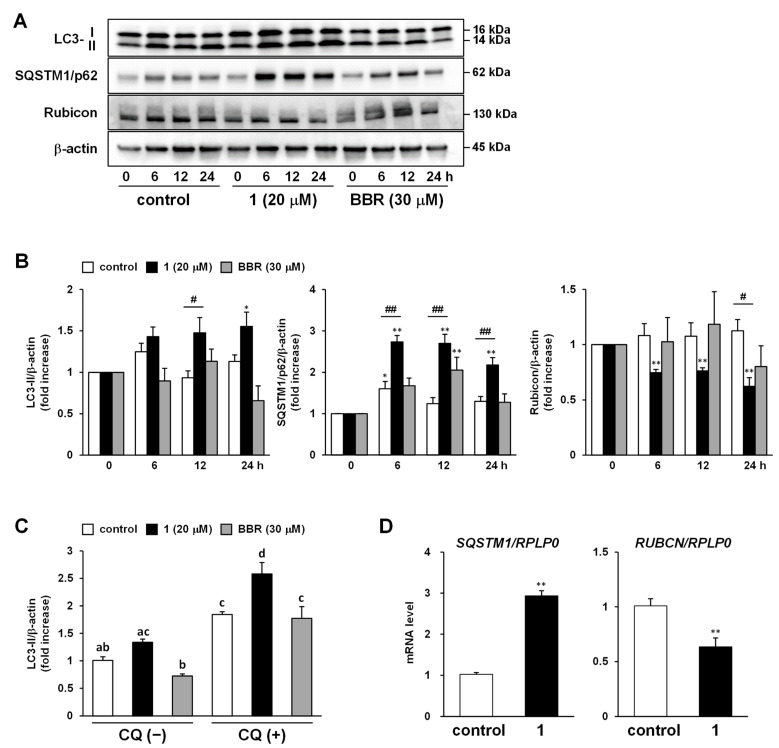
Expression of autophagy markers in HepG2 cells pretreated with high glucose and oleic acid concentrations. (**A**,**B**) Time-dependent expression of microtubule-associated protein 1 light chain 3 (LC3), sequestosome 1 (SQSTM1/p62), and Run domain Beclin-1 interacting and cysteine-rich containing protein (Rubicon) after application of 7-Deacetoxy-7-oxogedunin (DAOG; **1**) and berberine chloride (BBR) in high glucose and oleic acid-pretreated HepG2 cells, determined by western blot analysis. Unedited blots are shown in Appendix A. Each bar represents the mean with standard error of the mean (S.E.M.) (*n* = 4); * *p* < 0.05, ** *p* < 0.01 vs. initial (0 h) expression (Dunnett); # *p* < 0.05, ## *p* < 0.01 vs. control group at each sampling point (Dunnett); (**C**) Autophagy flux assay for compound **1** and BBR with or without chloroquine (CQ; 100 μM). CQ was added 6 h before harvesting cells. Each bar represents the mean with S.E.M. (*n* = 4); different letters indicate significant differences, *p* < 0.05 (Tukey–Kramer’s HSD); (**D**) mRNA expression levels of p62 and Rubicon analyzed by quantitative reverse transcription polymerase chain reaction, standardized using RPLP0. Cells were collected 4 h after the application of samples for mRNA extraction. Each bar represents the mean with S.E.M. (*n* = 4); ** *p* < 0.01 vs. control group (Student’s *t*).

**Table 1 ijms-23-13141-t001:** Effect of limonoids (**1**–**14**) on triglyceride (TG)/protein content in HepG2 cells.

Treatment	TG/Protein Content in the Homogenate (% of Control)		
	5 μM	10 μM	20 μM
Gedunin-type			
7-Deacetoxy-7-oxogedunin (DAOG; **1**)	87.5 ± 0.6 **	75.2 ± 0.7 **	70.0 ± 0.9 **
Gedunin (**2**)	91.7 ± 3.7 **	80.8 ± 0.9 **	74.0 ± 1.1 **
6α-Acetoxygedunin (**3**)	98.9 ± 1.6	101.6 ± 1.4	101.9 ± 1.2
Epoxyazadiradione (**4**)	85.3 ± 1.4 **	86.6 ± 1.6 **	73.4 ± 2.0 **
17β-Hydroxyazadiradione (**5**)	98.2 ± 2.7	92.1 ± 1.6*	79.9 ± 0.6 **
7-Deacetoxy-7α-hydroxygedunin (**6**)	81.3 ± 1.6 **	71.1 ± 0.7 **	61.0 ± 1.2 **
Andirolide (**7**)	95.0 ± 2.7	89.2 ± 2.8*	87.4 ± 2.2 **
6α-Hydroxygedunin (**8**)	95.0 ± 2.2	91.4 ± 1.1 **	84.5 ± 1.1 **
Andirobin-type			
Methyl angolensate (**9**)	114.7 ± 1.5 *	114.8 ± 0.7 *	118.6 ± 3.1 *
Obacunol-type			
Nomilin (**10**)	97.3 ± 2.8	101.2 ± 1.4	98.1 ± 0.6
Obacunone (**11**)	98.7 ± 2.9	96.0 ± 1.3	87.7 ± 1.9 **
Limonin (**12**)	102.2 ± 2.1	100.8 ± 0.3	94.6 ± 0.2
Nimbin-type			
Nimbin (**13**)	109.0 ± 4.7	106.1 ± 2.4	107.3 ± 0.9
Salannin-type			
Nimbolide (**14**)	74.6 ± 0.8 **	72.9 ± 1.2 **	140.9 ± 2.0 **^,†^
	**7.5 μM**	**15 μM**	**30 μM**
Berberine chloride (BBR)	91.9 ± 1.3 **	89.7 ± 0.9 **	84.1 ± 0.3 **
	**10 μM**	**30 μM**	**100 μM**
*Trans*-tiliroside	97.4 ± 1.4	82.6 ± 1.8 **	72.9 ± 1.0 **

Each value represents the mean ± standard error of the mean (S.E.M.) (*n* = 4). Significantly different from the control, * *p* <0.05, ** *p* <0.01 (Dunnett). The dagger sign (†) denotes cytotoxicity.

## Data Availability

The data supporting the findings of this study are available from the corresponding author upon reasonable request.

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
