# Peer review of "A Gedunin-Type Limonoid, 7-Deacetoxy-7-Oxogedunin, from Andiroba (Carapa guianensis Aublet) Reduced Intracellular Triglyceride Content and Enhanced Autophagy in HepG2 Cells"

_ijms, 2022, doi:10.3390/ijms232113141_

Round 1
Reviewer 1 Report
This paper reported the chemical studies on seed oil of Carapa guianensis. The inhibitory effects on glyceride accumulation of pure compounds were also investigated. The content and result are suitable for IJMS after minor revised.
1. I recommend author discuss some structure-activity relationship in different activity results.
2. The language of the manuscript needs extensive improvement there are too many indistinct expression in the paper. The article should be polished by native language expert.
3. The authors have concluded that " gedunin-type limonoids from Andiroba could ameliorate fatty liver by inducing autophagy" However, the evidence is not enough, please revise this sentence.
Author Response
We are grateful to your reviewing our manuscript and providing valuable suggestions to improve the manuscript. We have incorporated all your comments and suggestions in our revised manuscript. I hope this new manuscript is acceptable for publication in Int. J. Mol. Sci.
Reviewer #1
This paper reported the chemical studies on seed oil of Carapa guianensis. The inhibitory effects on glyceride accumulation of pure compounds were also investigated. The content and result are suitable for IJMS after minor revised.
- I recommend author discuss some structure-activity relationship in different activity results.
→ Thank you for your indication. We added following sentences discussing about structure-activity relationship.
Gedunin-type limonoids (1–8) are intact ABCD limonoid ring skeleton or the D-ring-seco limonoids, while compound 9is the B,D-ring-seco limonoid, compounds 10–12 are the A,D-ring-seco-limonoids, and compounds 13 and 14 are members of the C-ring-seco class (Figure 1). Therefore, the intact A, B, and C-rings of limonoid may be essential for the activity expression, however, further evaluation of wider range of limonoids is required. In the gedunin-type limonoids, 7-deacetoxy-7a-hydroxygedunin (6) with a hydroxy group at C-7 position showed the strongest activity, while the activity vanished in the case of 6a-acetoxygedunin (3) with two acetoxy groups at C-6 and C-7 positions. The activity was enhanced by the presence of an oxo-substituent at C-7 position. This indicates that the substitutions at these positions in the gedunin skeleton affected the activity.
Line 265-274.
- The language of the manuscript needs extensive improvement there are too many indistinct expression in the paper. The article should be polished by native language expert.
→ We have reviewed and revised the English wording throughout the manuscript again. In addition, we have been attached the certificate of English editing.
- The authors have concluded that " gedunin-type limonoids from Andiroba could ameliorate fatty liver by inducing autophagy" However, the evidence is not enough, please revise this sentence.
→ Following your instruction, we revised the sentence as follows.
These findings suggested that gedunin-type limonoids from Andiroba could ameliorate fatty liver, and that the action of DAOG in particular is mediated by autophagy.
Line 37-38.

Reviewer 2 Report
In this study, Nagatomo and colleagues aimed to address the ability of several limonoids, particularly that of gedunin-type limonoid 7-deacetoxy-7-oxogedunin in reducing the intracellular content of triglycerides and in enhancing autophagy in HepG2 cells. This is an interesting work with plausible applications in more specific studies and with in vivo models. Despite of the general quality and interest, there are several aspects that should be taken into consideration:
- The abstract section should be improved, namely adding some numerical data to represent how effective are the compounds studied.
- Introduction: 1) the authors should avoid to use personal expressions, like "our", etc.; 2) l.60-76: why no mention was done to ESPEN practical guidelines in the field (ESPEN practical guideline: Clinical nutrition in liver disease, https://www.espen.org/files/ESPEN-Guidelines/ESPEN_practical_guideline_Clinical_nutrition_in_liver_disease.pdf); 3) in addition, the pertinence of this study and relevance/difference over the already existing ones should be stated;
- Results section needs some numerical data so that readers can have a clearer idea on the effectiveness of the studied compounds
- Discussion needs to pay attention to the safety and pharmacokinetic aspects of the studied molecules; to what extent can it be used without triggering side effects or even toxicity to cells and in vivo models? also, it is important to perform additional experiments to address such extremely important aspects
- Materials and methods: 1) l. 351-373: this information is normally added when describing the procedures and assays used; please revise; 2) the oil of Carapa guianensis was obtained from where (l. 375-377)? was it deposited where? 3) DMSO (l. 416) was used at which concentration? above some doses it is toxic to cells; thus, how can authors infer about the absence of interference from this origin?
- Conclusion is extremely devoted to DAOG. Considering that the title of the manuscript also focused the same, what was the intent of also adding data about other compounds if no specific analysis and discussion was provided? please carefully revise the whole manuscript.
Author Response
We are grateful to your reviewing our manuscript and providing valuable suggestions to improve the manuscript. We have incorporated all your comments and suggestions in our revised manuscript. I hope this new manuscript is acceptable for publication in Int. J. Mol. Sci.
Reviewer #2
In this study, Nagatomo and colleagues aimed to address the ability of several limonoids, particularly that of gedunin-type limonoid 7-deacetoxy-7-oxogedunin in reducing the intracellular content of triglycerides and in enhancing autophagy in HepG2 cells. This is an interesting work with plausible applications in more specific studies and with in vivo models. Despite of the general quality and interest, there are several aspects that should be taken into consideration:
- The abstract section should be improved, namely adding some numerical data to represent how effective are the compounds studied.
→ As per your indication, we added numerical data in the abstract section as shown in Line 27-32.
The gedunin-type limonoids, such as DAOG (% of control at 20 mM: 70.9 ± 0.9%), gedunin (2, 74.0 ± 1.1%), epoxyazadiradione (4, 73.4 ± 2.0%), 17-hydroxyazadiradione (5, 79.9 ± 0.6%), 7-deacetoxy-7-hydroxygedunin (6, 61.0 ± 1.2%), andirolide H (7, 87.4 ± 2.2%), and 6-hydroxygedunin (8, 84.5 ± 1.1%), were observed to reduce the TG content at lower concentrations than berberine chloride (BBR, a positive control, 84.1 ± 0.3% at 30 mM) in HepG2 cells pretreated with high glucose and oleic acid.
- Introduction: 1) the authors should avoid to use personal expressions, like "our", etc.; 2) l.60-76: why no mention was done to ESPEN practical guidelines in the field (ESPEN practical guideline: Clinical nutrition in liver disease, https://www.espen.org/files/ESPEN-Guidelines/ESPEN_practical_guideline_Clinical_nutrition_in_liver_disease.pdf); 3) in addition, the pertinence of this study and relevance/difference over the already existing ones should be stated;
→ Thank you for your valuable indication.
1) We revised the sentences in the introduction section as shown in Line 57, 89, and 91.
2) The manuscript has been revised to mention the ESPEN guideline.
Line 68-70:
The first line of treatment for NAFLD/NASH coexisting with obesity as defined by the ESPEN guideline is intensive life style intervention leading to weight loss and exercise leading to hepatic fat reduction [26].
3) We have been added following sentences in the Introduction.
Line 78-82:
We have been searching for constituents from natural products that can contribute to the prevention and improvement of NAFLD or any other lifestyle-related disease in vivo models using mice or in cell-based assay using HepG2 cells. Studies on hepatic triglyceride metabolism often use HepG2 cells and other cell lines derived from liver as well as liver slices or primary hepatocytes [37].
- Results section needs some numerical data so that readers can have a clearer idea on the effectiveness of the studied compounds
→ As per your indication, we added numerical data in the Results section as shown in Line 146-150, 176-181, 188-200, 211-219, and 222-232.
Line 146-150:
The cell viability decreased in the presence of DAOG (1) at concentrations of more than 10 M (96.2 ± 0.5%), with no change in appearance (Figure 3). On the contrary, the cells were almost entirely killed in the presence of 100 µM BBR (1.6 ± 0.7%), while BBR at a concentration of 30 µM (96.2 ± 0.4%) showed little effects (all of numerical data described in Supplementary Material Table S1).
Line 176-181
Compound 1 (% of control: 71.3 ± 1.6% at 20 mM) and BBR (82.6 ± 2.4% at 30 mM) showed a concentration-dependent reduction of TG accumulated intracellularly using high glucose and oleic acid concentrations, as indicated by the blank bars in Figure 4A. In contrast, the activity of DAOG (1) and BBR was significantly attenuated by the concomitant use of compound C (Figure 4A, filled bars, 97.9 ± 0.9% at 20 mM of 1, 149.0 ± 2.6% at 30 mM of BBR) as fully described in Table S2.
Line 188-200
Figures 4B and C indicate that BBR markedly increased the p-AMPK/AMPK ratio (3.21 ± 0.66 fold increase, Table S3). In addition, compound 1 also tended to increase the p-AMPK/AMPK ratio (1.43 ± 0.07 fold increase, Table S3); however, the change was in-significant relative to the control (only vehicle-treated). Acetyl-CoA carboxylase (ACC) is a key molecule in fatty acid metabolism and is negatively regulated by AMPK. ACC catalyzes the conversion of acetyl-CoA to malonyl-CoA, which inhibits carnitine palmitoyltransferase (CPT)-1. Because CPT-1 catalyzes the transfer of the fatty acid groups from acyl-CoA to carnitine and increases their transport from cytosol to mitochondria, inactivation of ACC by AMPK enhances fatty acid degradation via -oxidation [59]. In our study, ACC phosphorylation was not clearly observed despite AMPK activation (0.86 ± 0.09 fold increase for 1, 0.96 ± 0.14 fold increase for BBR, Table S3). However, p-ACC/ACC ratio significantly decreased due to treatment with compound C (Table S3).
Line 211-219:
Compound 1 stimulated the conversion of LC3-I into LC3-II, and the LC3-II expression at 24 h after the addition of compound 1 (1.55 ± 0.17 fold increase, Table S4) was significantly increased compared to the initial value. SQSTM1/p62 is an autophagy adaptor that directly binds to LC3 to induce autophagosome formation [60]. Typically, SQSTM1/p62 is degraded during autophagy, but our results showed that compound 1 caused a rapid increase in SQSTM1/p62 protein expression (2.74 ± 0.15 fold increase at 6 h, Table S4), followed by a decrease over time. Meanwhile, the expression of Rubicon, a negative regulator of autophagy [61], was observed to decrease in response to treatment with compound 1 (0.62 ± 0.08 fold increase at 24 h, Table S4).
Line 222-232:
Chloroquine hydrochloride (CQ) inhibits autophagy at its final stage by suppressing the fusion of autophagosomes with lysosomes [62]. The results in Figure 5C showed that LC3-II expression in the control group was significantly increased in the presence of CQ (1.84 ± 0.05 fold increase, Table S5); however, even more significant increase was observed with the combination of compound 1 and CQ (2.58 ± 0.21 fold increase, Table S5). Furthermore, cells were collected 4 h after sample addition, and mRNA expression was confirmed by real-time quantitative reverse transcription polymerase chain reaction (qRT-PCR); the mRNA expression levels of SQSTM1 and RUBCN were observed to increase and decrease, respectively, compared to the control (relative mRNA level was 2.93 ± 0.07 for SQSTM1, 0.63 ± 0.02 for RUBCN, Table S6).
- Discussion needs to pay attention to the safety and pharmacokinetic aspects of the studied molecules; to what extent can it be used without triggering side effects or even toxicity to cells and in vivo models? also, it is important to perform additional experiments to address such extremely important aspects
→ Thank you for your important suggestion. Below sentences have been added in the end of Discussion section.
Line 365-372
In the current study, only in vitro evaluation using cells was conducted. The pharmacokinetics of limonoids have been well studied, especially for limonin (12). Generally, the bioavailability of limonin is very low due to its low solubility and poor permeability [107,108]. Although there are no reports on the pharmacokinetics of compound 1, it is also sparingly soluble in water, so the bioavailability is thought to be low. However, Matsumoto et al. elucidated the anti-obesity and the anti-diabetic effects of DAOG (1) by in vivo model using high-fat-diet-fed mice [71]. Thus, compound 1 is expected to exhibit oral absorbability to some extent.
- Materials and methods: 1) l. 351-373: this information is normally added when describing the procedures and assays used; please revise; 2) the oil of Carapa guianensis was obtained from where (l. 375-377)? was it deposited where? 3) DMSO (l. 416) was used at which concentration? above some doses it is toxic to cells; thus, how can authors infer about the absence of interference from this origin?
→ Thank you for your instructions.
- Materials and methods have been revised as shown in Line 375-394. The General section describes the experimental devices for the isolation and the identification of constituents from guianensis. Papers in similar fields, including our previous reports, describe the apparatuses in the same way.
- We have inserted the Plant material section and described the detail of the oil from guianensis as shown in Line 395-399.
Line 395-399
4.2. Plant material
The flower and seed oils of C. guianensis Aublet (Meliaceae) were collected in Amazon, Brazil, in March of 2006, 2011, and 2013. Voucher specimens (CG-01-1, CGS-01-1, and CGS-01-2) were deposited at the Herbarium of the Laboratory of Medicinal Chemistry, Osaka University of Pharmaceutical Sciences, as described previously [16].
- DMSO was used up to 1% as described in Line 441-442. We confirmed in advance that this concentration would not affect the test system and the cell integrity.
- Conclusion is extremely devoted to DAOG. Considering that the title of the manuscript also focused the same, what was the intent of also adding data about other compounds if no specific analysis and discussion was provided? please carefully revise the whole manuscript.
→ Thank you for your valuable suggestions. One of our purposes of this study is to characterize the biofunction of limonoids isolated from Andiroba, so we demonstrated and compared the lipid-reducing effect of several types of limonoids. We have added sentences of the structure-activity relationships of limonoids to the Discussion and Conclusion for further understanding as follows;
Line 265-274:
Gedunin-type limonoids (1–8) are intact ABCD limonoid ring skeleton or the D-ring-seco limonoids, while compound 9is the B,D-ring-seco limonoid, compounds 10–12 are the A,D-ring-seco-limonoids, and compounds 13 and 14 are members of the C-ring-seco class (Figure 1). Therefore, the intact A, B, and C-rings of limonoid may be essential for the activity expression, however, further evaluation of wider range of limonoids is required. In the gedunin-type limonoids, 7-deacetoxy-7a-hydroxygedunin (6) with a hydroxy group at C-7 position showed the strongest activity, while the activity vanished in the case of 6a-acetoxygedunin (3) with two acetoxy groups at C-6 and C-7 positions. The activity was enhanced by the presence of an oxo-substituent at C-7 position. This indicates that the substitutions at these positions in the gedunin skeleton affected the activity.
Line 506-513.
Several types of limonoids, including Andiroba (C. guianensis)-derived limonoids, were examined for their intracellular TG-reducing effects, and we found that the gedunin-type limonoids characteristically reduced intracellular TG content. Consideration of the structure-activity relationship suggested that the substituents at the C-6 and C-7 positions in the gedunin skeleton affected the activity. DAOG (1), a principal limonoid of Andiroba, was observed to be involved in AMPK activation as part of its mechanism of action. In addition, compound 1 increased autophagy markers and decreased the expression of autophagy suppressors, suggesting that its action is mediated by autophagy.

Reviewer 3 Report
In this manuscript, Nagatomo et al. investigated the intracellular TG-reducing effects of several types of limonoids, including Andiroba (C. guianensis). The authors found that the gedunin-type limonoids attenuated intracellular TG content. Furthermore, they indicated that DAOG was involved in AMPK activation and increased autophagy markers, such as LC3 and p62. This is very interesting paper, because these results could be helpful to develop the potential preventive agents for fatty liver using gedunin-type limonoids. Thus, I recommend the publication of this manuscript for “International Journal of Molecular Science.
Author Response
We are grateful to your reviewing our manuscript and providing valuable suggestions to improve the manuscript. We have incorporated all your comments and suggestions in our revised manuscript. I hope this new manuscript is acceptable for publication in Int. J. Mol. Sci.
Reviewer #3
In this manuscript, Nagatomo et al. investigated the intracellular TG-reducing effects of several types of limonoids, including Andiroba (C. guianensis). The authors found that the gedunin-type limonoids attenuated intracellular TG content. Furthermore, they indicated that DAOG was involved in AMPK activation and increased autophagy markers, such as LC3 and p62. This is very interesting paper, because these results could be helpful to develop the potential preventive agents for fatty liver using gedunin-type limonoids. Thus, I recommend the publication of this manuscript for “International Journal of Molecular Science.
→ Thank you for your careful peer review. The manuscript has been revised according to other reviewers’ comments.

Round 2
Reviewer 2 Report
All comments raised were properly addressed.